# Resourceful Treatment of Battery Recycling Wastewater Containing H_2_SO_4_ and NiSO_4_ by Diffusion Dialysis and Electrodialysis

**DOI:** 10.3390/membranes13060570

**Published:** 2023-05-31

**Authors:** Sifan Wu, Haitao Zhu, Yaqin Wu, Shuna Li, Gaoqi Zhang, Zhiwei Miao

**Affiliations:** 1Hangzhou Water Treatment Technology Development Center Co., Ltd., Hangzhou 310012, China; wusifan@chinawatertech.com (S.W.); wuyq@chinawatertech.com (Y.W.); lishuna@chinawatertech.com (S.L.); zhanggaoqi@chinawatertech.com (G.Z.); miaozhiwei@chinawatertech.com (Z.M.); 2Zhejiang Key Laboratory of Seawater Desalination Technology, Hangzhou 310012, China; 3Bluestar (Hangzhou) Membrane Industries Co., Ltd., Hangzhou 311103, China

**Keywords:** battery wastewater, diffusion dialysis (DD), electrodialysis (ED), recycling

## Abstract

Facing the increasing demand for batteries worldwide, recycling waste lithium batteries has become one of the important ways to address the problem. However, this process generates a large amount of wastewater which contains high concentration of heavy metals and acids. Deploying lithium battery recycling would cause severe environmental hazards, would pose risks to human health, and would also be a waste of resources. In this paper, a combined process of diffusion dialysis (DD) and electrodialysis (ED) is proposed to separate, recover, and utilize Ni^2+^ and H_2_SO_4_ in the wastewater. In the DD process, the acid recovery rate and Ni^2+^ rejection rate could reach 75.96% and 97.31%, respectively, with a flow rate of 300 L/h and a W/A flow rate ratio of 1:1. In the ED process, the recovered acid from DD is concentrated from 43.1 g/L to 150.2 g/L H_2_SO_4_ by the two-stage ED, which could be used in the front-end procedure of battery recycling process. In conclusion, a promising method for the treatment of battery wastewater which achieved the recycling and utilization of Ni^2+^ and H_2_SO_4_ was proposed and proved to have industrial application prospects.

## 1. Introduction

In recent years, the rapid development of the electric vehicle industry has led to an increasing demand for batteries. Due to their long storage life, high energy density, low self-discharge rate, small size, and light weight, lithium-ion batteries (LIBs) are widely used in electronic devices [1,2]. However, lithium resources are limited and might face shortages in the near future [3,4]; therefore, it is imperative to recycle postconsumer LIBs effectively [5]. In the battery recycling process, acid solution is used to leach valuable metals [6], resulting in wastewater with high concentrations of acid and heavy metals [7,8], which would cause severe environmental hazards and pose risks to human health while also leading to resource waste [9]. Therefore, recycling and utilizing the acids and heavy metals in wastewater can not only protect the environment effectively but also save resources and generate significant economic benefits.

To manage the wastewater of the battery recycling industry, several treatment methods can be used, including chemical precipitation [10], extraction [11,12,13], electrocoagulation [14], ion exchange [15], and membrane separation [16,17,18]. Among these methods, the membrane process has drawn more attention in recent years and is mentioned as a simple, environmental, and effective method with applications in water treatment, desalination, the pharmaceutical industry, and other fields. As one of the membrane technologies, diffusion dialysis (DD) is an especially attractive method that is widely used in the treatment of acidic wastewater, which has high proton transmittance, a high rejection rate of heavy metals, and low energy consumption and cost [19]. In some studies, DD was used to separate and recover mineral acids and transition metals from electroplating industry wastewater, for separating H_2_SO_4_ and CuSO_4_ [20], or for separating acids and heavy metals such as iron and zinc from pickling water [21]. Diffusion dialysis technology is based on anion exchange membranes (AEMs), separating the acid and metal ions by a difference in concentration between two compartments. The acid group anions can be freely transported through the AEM, while the metallic cations are rejected by the positive fixed charges of the AEM [22,23]. However, protons can migrate through the membrane due to their small dimension and the tunneling mechanism [21], after which they combine with the acid group anions. Therefore, DD is widely used in acid separation and recovery [22,23].

As another membrane technology, electrodialysis (ED) is always used in desalination, concentration, and separation [24] due to its low energy consumption, low environmental impact, and low operational costs [25,26]. ED uses anion exchange membranes with positively charged functional groups and cation exchange membranes with negatively charged functional groups, which selectively pass counter-ions with opposite charges and block co-ions with the same charge sign [27]. For this reason, ED can be used in acid concentration to obtain acid solutions of high concentration. Thus, DD followed by ED processing could solve the problem of the low concentration of acid solution produced by DD, concentrating the acid solution of the DD process.

In this study, DD and ED technologies were combined to deal with battery recycling wastewater for acid and metal recovery. DD was firstly used to separate the acid and heavy metals, then the recovered acid solution was concentrated through two-stage ED. The concentrate of the ED could be reused in the front-end process of the battery recycling industry, and the diluate of ED could be reused as the diffusate of DD. In our study, pilot scale experiments were tested to verify the feasibility of this process and proved to have a promising application prospect.

## 2. Materials and Methods

### 2.1. Materials

The wastewater from the battery recycling industry was used as the feed water in the DD process of this work, mainly containing H_2_SO_4_ and NiSO_4_. The components of the feed water are shown in Table 1.

The DD stack was formed by 100 AEMs (HWTT^®^ DD-6, Hangzhou, China), each of them measuring 0.10 mm, and every membrane was separated by a spacer with a thickness of 1.0 mm. The characteristics of the membrane are shown in Table 2. The unit had a total active membrane area of 60.5 m^2^ and a projected dimension of 55 × 110 cm. The feed wastewater was filled in the diffusate cell, and the diluate of ED was filled in the dialysate cell. A diagram of the DD set-up and membrane stack is shown in Figure 1a.

The ED stack of the first stage was formed by 100 cell pairs, which included 101 sheets of CEMs (LANCYTOM^®^ CT-4, Hangzhou, China) and 100 sheets of AEMs (HWTT^®^ A2N). Each membrane was separated by a spacer with a thickness of 1.0 mm. The unit had a total active membrane area of 121.605 m^2^ and a projected dimension of 55 × 110 cm. The second stage ED included 101 sheets of CEMs (LANCYTOM^®^ CT-4) and 100 sheets of AEMs (LANCYTOM^®^ ATD), with a total active membrane area of 36.18 m^2^ and a projected dimension of 30 × 60 cm. The characteristics of the membranes are shown in Table 2. The membranes were purchased from Lanran Co., Hangzhou, China. The electric current was supplied by a DC stabilized power supply (DT-120P, Dongyang Datong Electrical Instrument Ltd., Datong, China), and the voltage and current could be directly read from the monitor. The current density of the ED process was kept constant at 250 A/m^2^. In the first stage of ED, the flow rates of dilution chamber and concentrate chamber were maintained at 1600 and 2400 L/h (ca. 1.5 and 2.3 cm/s). In the second stage of ED, the flow rates were maintained at 800 and 1200 L/h (ca. 1.3 and 2.0 cm/s). The diagram of the ED set-up and membrane stack is shown in Figure 1b.

### 2.2. Analysis

The concentration of H_2_SO_4_ was determined by titration with a 0.5 mol/L NaOH standard solution with bromocresol green-methyl red as an indicator. The concentration of Ni^2+^ was measured by ICP-OES (Agilent 720ES, Santa Clara, CA, USA). The solution conductivity was measured by a conductivity meter (S220 type, Mettler-Toledo, Greifensee, Switzerland).

In the DD process, the acid recovery ratio (*R*, %) was calculated according to Equation (1) [28]:(1)$$R=\frac{C_{R}Q_{R}}{C_{F_{0}}Q_{F}}\times 100\%$$
where *C_R_* is the H_2_SO_4_ concentration of the recovered acid (mol/L), *Q_R_* is the flow rate of the recovered acid (L/h), *C_F_*_0_ is the initial H_2_SO_4_ concentration of the feed (mol/L), and *Q_F_* is the flow rate of the feed (L/h).

The rejection ratio of Ni (*r*, %) was calculated according to Equation (2):(2)$$r=(1-\frac{c_{R}Q_{R}}{c_{F_{0}}Q_{F}})\times 100\%$$
where *c_R_* is the Ni^2+^ concentration of the recovered acid (mol/L), *Q_R_* is the flow rate of the recovered acid (L/h), *c_F_*_0_ is the initial Ni^2+^ concentration of the feed (mol/L), and *Q_F_* is the flow rate of the feed (L/h).

Current efficiency (*η*, %) and energy consumption (*E*, kWh/kg H_2_SO_4_) were important to evaluate the performance of the ED process, and were calculated according to Equations (3) and (4), respectively [29].
(3)$$\eta =\frac{\left(c_{t}-c_{0}\right)V_{t}F}{NIt}\times 100\%$$
where *η* is the current efficiency (%), *C_t_* is the concentration of H^+^ in concentration chamber at time *t* (mol/L), *C*_0_ is the concentration of H^+^ in concentration chamber at time 0 (mol/L), *V_t_* is the volume of the concentrated acid solution received at time *t* (L), *F* is the Faraday’s constant (96,485 C/mol), *N* is the number of repeating units of the stack (*N* = 100), *I* is the applied current (A), and *t* is the operation time (s).
(4)$$E=\int_{0}^{t}\frac{UI}{c_{t}V_{t}M_{b}}dt$$

Here, *E* is the energy consumption (kWh/kg H_2_SO_4_), *U* is the applied voltage (V), *I* is the applied current (A), *C_t_* is the concentration of H_2_SO_4_ solution at time *t* (mol/L), *V_t_* (L) is the volume of the concentrated acid solution received at time *t*, and *M_b_* is the molar weight of H_2_SO_4_ (98 g/mol).

## 3. Results

### 3.1. Diffusion Dialysis Process

Diffusion dialysis was used to separate the acid and heavy metals in the feed wastewater, and some parameters were explored for the optimal conditions. The feed water was filtered by a self-made plate filter with a pore size of 0.1 μm, made by PTFE. The particulate matters were rejected to ensure the stable operation of the DD and ED processes.

#### 3.1.1. Effect of Flow Rate

As one of the salient parameters of the diffusion dialysis performance, the flow rate was investigated in this section. The flow rate ratio of water to feed (W/F) was maintained at 1:1, with the feed flow rate increased from 150 L/h to 350 L/h. The acid recovery rate and rejection rate of Ni^2+^ is shown in Figure 2. It was clearly seen that the acid recovery rate decreased from 87.62% to 73.92% as the feed flow rate increased from 150 L/h to 350 L/h, which was due to the short retention time for acid adsorption in the membrane surfaces. There was not enough time for acid to transport through the membranes as the feed flow rate increased [30,31]. On the other hand, the rejection rate of Ni^2+^ increased from 94.35% to 97.43% as the feed flow rate increased from 150 L/h to 350 L/h, because the higher flow rate shortened the retention time for Ni^2+^ to form complex ions and reduced the opportunity for Ni^2+^ to diffuse across the membrane, which was similar to other studies [22,23,32]. With these considerations in mind, the feed flow rate was selected as 300 L/h for following experiments.

#### 3.1.2. Effect of the Flow Rate Ratio

To investigate the effect of the flow rate ratio of water to feed (W/F) on the DD performance, five experimental runs were conducted, varying the flow rate ratio to 0.5, 0.75, 1.0, 1.25, and 1.5 while the feed flow rate was maintained at 300 L/h. Figure 3 shows that the acid recovery rate increased from 70.12% to 77.89% when the W/F flow rate ratio increased from 0.5 to 1.5. This was because the stagnant layer would become thinner while the diffusate flow rate increased. The concentration difference between the membrane–diffusate interface and the bulk of the diffusate tended to be constant as the feed remained unchanged; however, the concentration gradient increased as a result of the decrease in the thickness of the stagnant layer. Therefore, the mass transfer by diffusion was improved so that more acid was transported through the membrane as the flow rate ratio increased [30,33]. Meanwhile, more Ni^2+^ also transported through the membrane, so that the rejection rate of Ni^2+^ decreased from 97.87% to 94.38% when the W/F flow rate increased from 0.5 to 1.5. As a result, to achieve high acid recovery and Ni^2+^ rejection, the W/F flow rate was selected as 1.0, while the feed flow rate and water flow rate were both 300 L/h; under these conditions, the acid recovery rate was 75.96% and the Ni^2+^ rejection rate was 97.31%.

### 3.2. Electrodialysis Process

The concentration of acid solution produced in DD process was not high enough to reuse in the front process, which always used in resin regeneration and cleaning of the pretreatment device, such as ultra-filtration, sand-bed filter, electrocoagulation, However, ED was used in this work to reach high concentrations of acid solution for cyclic utilization in the front process. The acid enrichment by ED was limited by some factors, such as proton leakage, acid back diffusion, and concentration polarization [34,35]. Therefore, it was difficult to obtain high-concentration acid solution through one-stage ED due to the high water transport at the high concentration gradient between diluted and concentrated solutions [36]. Two-stage ED was used in this work for higher acid concentration, and achieved lower energy consumption and lower costs. The diagram of two-stage ED is shown in Figure 4.

#### 3.2.1. The First Stage of ED

The recovered acid of the DD process was used as the diluate in the first stage of ED, for which the H_2_SO_4_ concentration was around 40 g/L. When the concentration decreased to 5 g/L, it could be reused as the diffusate in DD to achieve the process cycle. As shown in Figure 4, the concentrate in the first stage of ED was used as the diluate in the second stage of ED. At the beginning, 0.1 mol/L H_2_SO_4_ was used as the concentrate solution to increase the initial reaction rate. Additionally, 0.2 mol/L H_2_SO_4_ was used as the electrode rinse. The initial volumes of the dilution tank and intermediate tank were 240 L and 120 L, respectively, and the current density was kept constant at 250 A/m^2^. It can be seen in Figure 5 that at 25 min the concentration of H_2_SO_4_ in dilution tank decreased from 43.1 g/L to 3.9 g/L; most of this went back to the DD diffusate, while the recovered acid was supplied to the dilution tank. Finally, the concentration of H_2_SO_4_ reached 80.4 g/L with an almost constant concentration in the concentration chamber at 100 min. Because of proton leakage, acid back diffusion, and water diffusion, which limited the concentration of the acid solution in the ED process, at the end of the experiment, the concentration of H_2_SO_4_ was almost unchanged. Therefore, the concentration of sulfuric acid can reach 80 g/L in the first stage of ED, and the diluate can be reused in the DD process as the diffusate.

#### 3.2.2. The Second Stage of ED

The first stage of ED was operated for pre-concentrated acid with high current efficiency and low energy consumption, while the second stage of ED was operated for a higher concentration of acid solution. Thus, the concentrate solution of the first stage of ED, which was around 80 g/L H_2_SO_4_, was continuously used as the diluate in the second stage of ED. Additionally, the initial concentrate solution was replaced with the concentration of the first stage of ED. It is shown in Figure 6 that the H_2_SO_4_ concentration of the dilution chamber stayed almost constant, while the H_2_SO_4_ concentration of the concentration chamber gradually increased and reached 150.2 g/L at 180 min with an almost constant concentration. The concentration was about twice that of the first stage of ED. The achievement of high H_2_SO_4_ concentration was possible because the membrane used in the second stage of ED has stronger resistance to hydrogen permeation, which can limit the proton leakage phenomenon due to its specific base materials and weakly basic exchange groups.

The current efficiency and energy consumption of the ED processes are shown in Figure 7. The current efficiency was in a low range of around 33% to 37% because of the back diffusion of the acid and the proton leakage. Similar results were also shown in other studies [37,38]. Due to the higher H_2_SO_4_ concentration combined with the larger concentration gradient, the current efficiency of the second stage of ED was lower than that of the first stage of ED and the energy consumption of the second stage of ED was higher than that of the first stage of ED. The total energy consumption of the ED processes was 1158 kWh/t H_2_SO_4_, as shown in Figure 7.

## 4. Conclusions

In this study, a coupling process of diffusion dialysis and electrodialysis was proposed to treat wastewater from the battery recycling industry to recover and concentrate valuable metals and acids. Firstly, the DD process was used to separate acid and heavy metals. With a flow rate of 300 L/h and flow rate ratio of 1:1, the acid recovery and Ni^2+^ rejection could reach 75.96% and 97.31%, respectively. With the high rejection rate of Ni^2+^, the nickel-rich solution could be used in battery production process. Secondly, the recovered acid was concentrated by two-stage ED, which concentrated the acid solution from 43.1 g/L to 150.2 g/L H_2_SO_4_. The diluate of ED was used as the diffusate for the DD process, and the concentrate was used in the front-end procedure of the battery recycling process. Consequently, a promising method for the treatment of battery wastewater was proposed, which recovered and utilized the Ni^2+^ and H_2_SO_4_ in the wastewater.

## Figures and Tables

**Figure 1 membranes-13-00570-f001:**
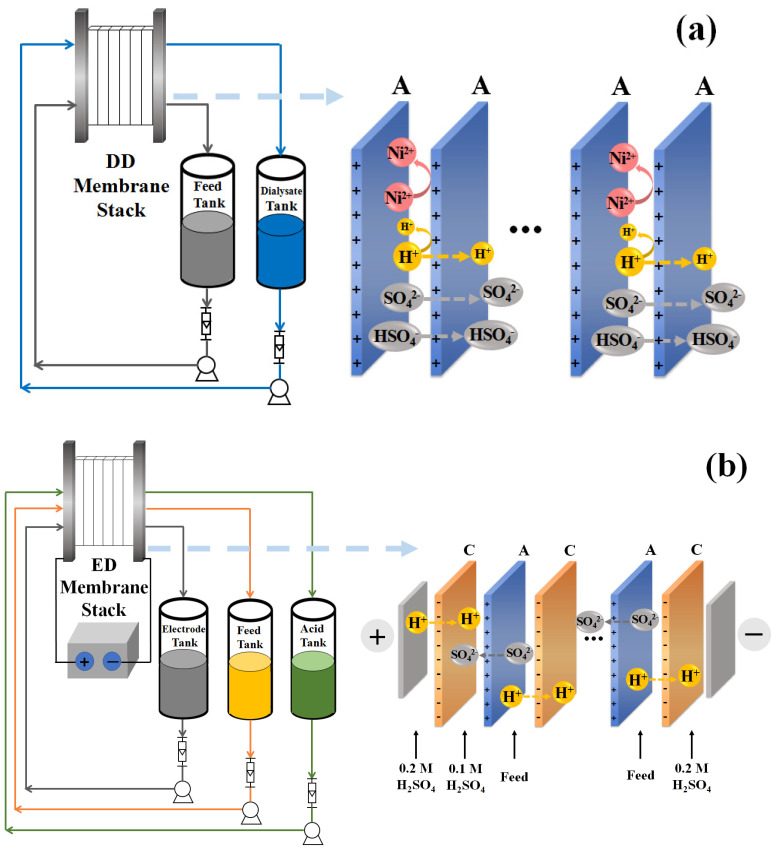
The diagram of (**a**) the DD set-up and membrane stack; (**b**) the ED set-up and membrane stack.

**Figure 2 membranes-13-00570-f002:**
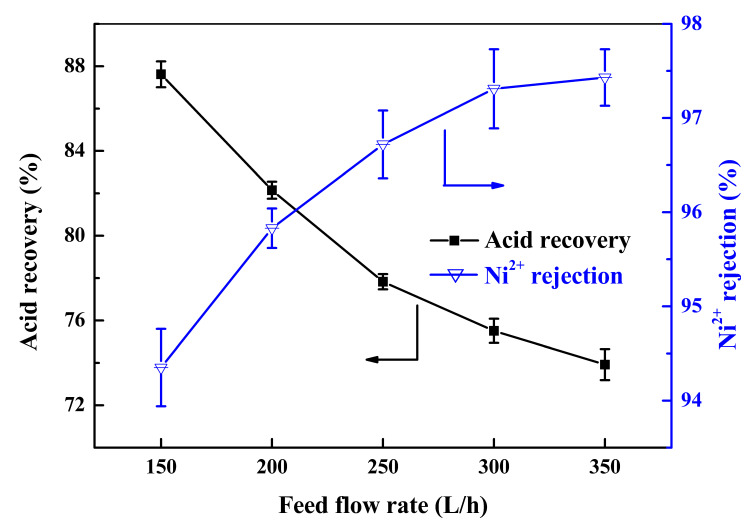
Effect of the feed flow rate on acid recovery and Ni^2+^ rejection.

**Figure 3 membranes-13-00570-f003:**
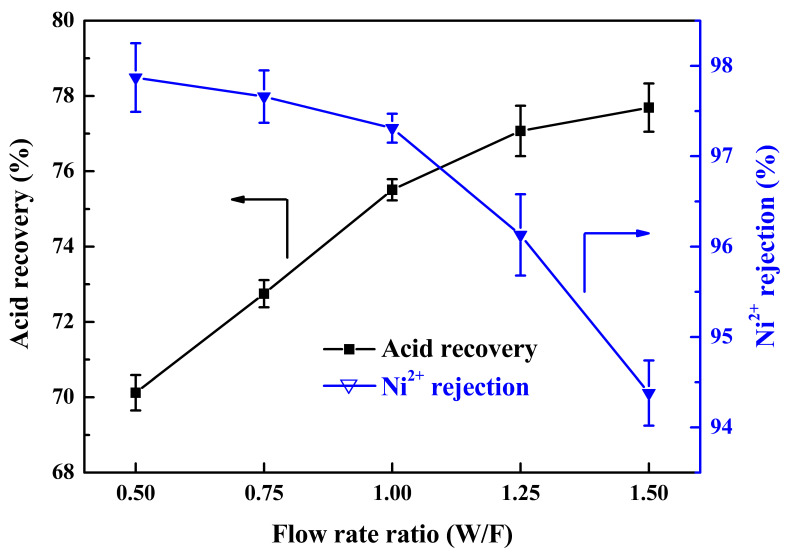
Effect of the flow rate ratio (W/F) on acid recovery and Ni^2+^ rejection.

**Figure 4 membranes-13-00570-f004:**
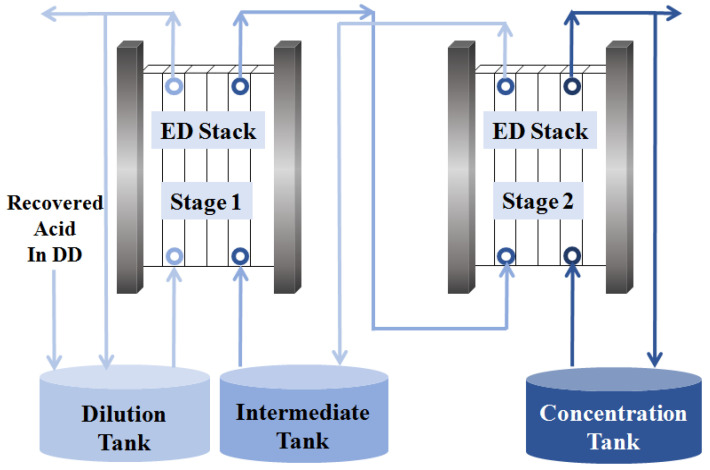
The diagram of two-stage ED.

**Figure 5 membranes-13-00570-f005:**
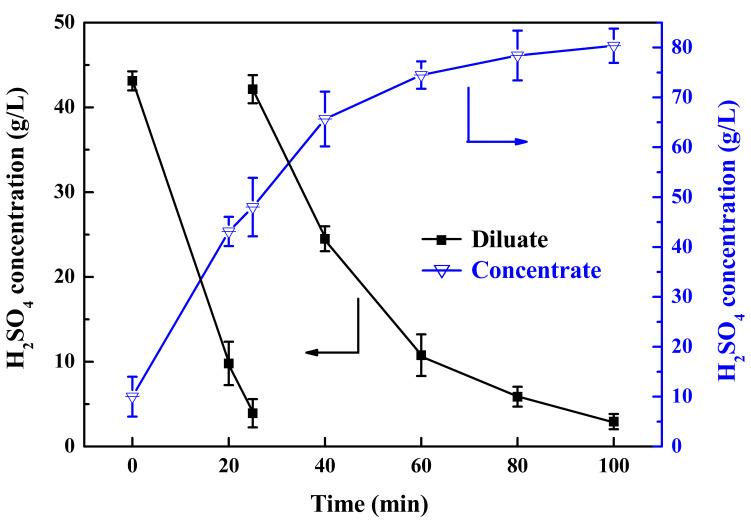
The concentrations of H_2_SO_4_ in diluate and concentrate in the first stage ED.

**Figure 6 membranes-13-00570-f006:**
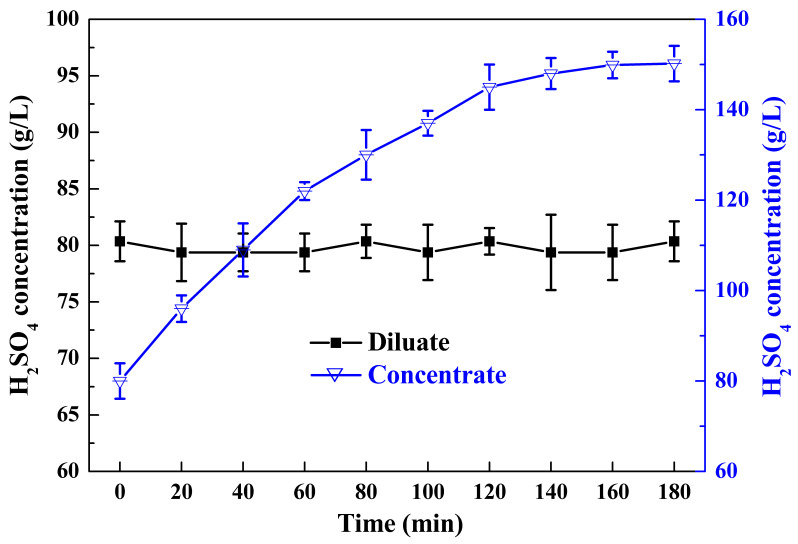
The concentrations of H_2_SO_4_ in diluate and concentrate of the second stage of ED.

**Figure 7 membranes-13-00570-f007:**
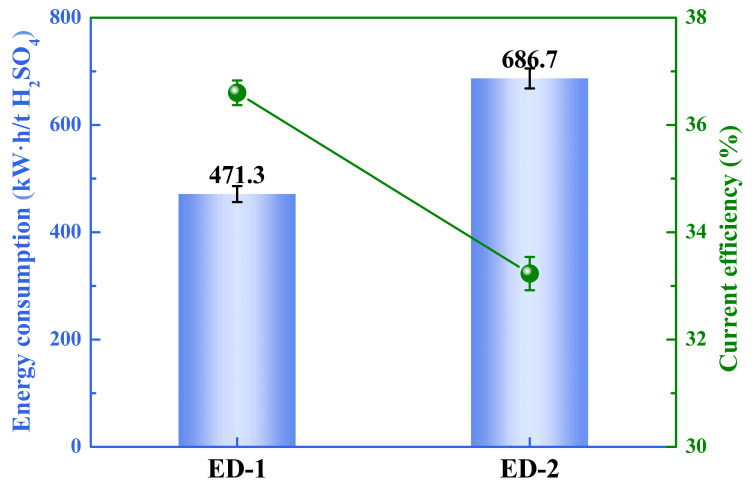
The energy consumption and current efficiency of the first and second stages of ED.

**Table 1 membranes-13-00570-t001:** Components of the feed.

Component	Concentration (g/L)
Ni	46.85
Fe	1.74
H_2_SO_4_	57.33
H_3_PO_4_	1.45

**Table 2 membranes-13-00570-t002:** Main properties of membranes used in DD and ED.

Membrane	Thickness (μm)	Area Resistance * (Ω·cm^2^)	Burst Strength (MPa)	pH	Tolerable Temperature (°C)	Used in
HWTT^®^ DD-6	100	1.0	≥0.25	0–14	10–40	DD
HWTT^®^ A2N	100	3.0	≥0.25	0–10	10–40	ED-1
LANCYTOM^®^ CT-4	100	3.6	≥0.50	0–14	25–40	ED-1/2
LANCYTOM^®^ ATD	160	4.3	≥0.60	0–4	25–40	ED-2

* The surface area resistance was measured with a 0.5 N–NaCl solution, at 25 °C.

## Data Availability

Data presented in this study can be available by contacting wusifan@chinawatertech.com.

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
