# Peer review of "Resourceful Treatment of Battery Recycling Wastewater Containing H2SO4 and NiSO4 by Diffusion Dialysis and Electrodialysis"

_membranes, 2023, doi:10.3390/membranes13060570_

Round 1

Reviewer 1 Report

The manuscript contains some interesting experimental results that deserve publication. However, before that, the manuscript should be significantly improved.

 Below, there are some comments.

 Please, calculate what are the losses of acid and salt during the proposed treatment (after each step, and the total losses).

 „the proton can migrate through the membrane due to its small dimension and the tunneling mechanism …” – any reference or explanation to „the tunneling mechanism”?

 „DD was widely used in acid separation and recovery in recent years ..” – recent?? Ca. 40 years ago the DD came into use.

 „membrane was separated by a blocker” – blocker (??) or spacer?

 Table 2. Main properties… - temperature of what?

 Figure 1a – it is not drawn that H+ and SO42- can also pass through the left AEM. What about HSO4- anions?

 Fig 1b – H2O is denoted as the input liquid. According to the statement in the Introduction:

„DD was firstly used to separate the acid and heavy metals, then the recovered acid solution was concentrated by the two-stage ED”

one would expect here the „recovered acid solution” from DD.

 „where cR is the Ni2+ concentration of (mol/L), QR is the flow rate of (L/h),” – of which solution?

 „Vt is the volume of the concentration chamber at time t (L)” – does the volume of the chamber changes with time?? This is the volume of the concentrated acid solution received at time t, right?

Eq.(3) applies for the I = const condition. It is not mentioned in the text whether it was a galvanostatic or potentiostatic mode of ED.

“the feed flow rate increased from 150 L/h to 350 L/h.” – what is the linear velocity of the feed flow?

“at higher feed flow rates, the diffusion layer at the surface of AEM was relatively thinner while the turbulence in the bulk solution was rather greater, which would hinder H+ to cross the membrane.” – please, explain. One would expect that the thinner the diffusion layer, the higher permeation because of a higher concentration difference across the membrane.

 “the higher flow rate would shorten the retention time for Ni2+ to form complex ions and reduce the opportunity for Ni2+ to diffuse across the membrane” – what complex ions does Ni2+ form? What is the stability constant of these ions?

 Figure 4. “The diagram of two-stage ED” – the inlet is lacking.

 Figure 5. – The description of curves is lacking.

 “the membrane used in the second stage ED has stronger hydrogen resistance.” – “hydrogen resistance” is too short - it should be named differently.

 “kW•h” – kWh

 Figure 6. “The concentrations of H2SO4 in dilution and concentration of the second stage ED.” – the description is insufficient: what is the blue line, black line?

 “the H2SO4 concentration of dilution chamber almost stayed constant,” – please, explain, why.

 Conclusions

“This section is not mandatory but can be added to the manuscript if the discussion is unusually long or complex.” – it should be removed.

Please, rewrite the conclusions. This text cannot be regarded as a conclusion.

 The quality of drawings should be improved (now they are blurry).

English should be improved, here are a few examples:

„Diffusion dialysis technology was based on anion …” – is based.

„The acid group anions can freely transported through the AEM …” – be.

„which can coupled with DD…” – be.

“was shown in Figure …” – is shown.

“The result was shown in Figure 3 that the acid recovery rate increased from 70.12% to 77.89% since the W/F flow rate ratio increased from 0.5 to 1.5.“ – it should be changed.

Author Response

Thanks for you comments.

Reviewer 2 Report

The paper describes a DD coupled with a two-stage ED process to recover acid and metals from a waste stream of an automotive battery recycling process. Despite the importance of the topic and the evident need to improve processes that point towards zero liquid discharge, I believe the article needs structural improvements to be published. The article is more focused on the process than on fundamental discussions. Nonetheless, it is notable that the authors have relevant results and scientific experience. However, some parts of the text are somewhat contradictory, and others do not have sufficient scientific depth. Below are minor and major changes that need to be reviewed and corrected where necessary.

• An English review is required to make the text more scientific and more fluid.

• In line 20 of the abstract, the authors report that they describe a method that achieved recycling and the reuse of acid and nickel, but throughout the manuscript, the reuse of the DD retentate, which would be rich in nickel, is not addressed.

• At the beginning of the introduction, the authors focus on the demand for lithium, but the treated waste stream is rich in nickel. To make the idea more coherent, the authors can provide more details about the waste stream. Is it real or synthetic? Did this waste stream, whether real or synthetic, come from a stage of recycling in which other elements were previously recovered?

• In the methodology, the materials are adequately described, but there is no scientific description of the method. Some examples (but not limited to) are: definition of the process parameters that were studied, how the applied current was obtained in the ED stages, instrumentation and control (monitoring of pH, conductivity, stack potential?).

• All figures have poor quality.

• Many elements that should be present in the methodology are presented in the results and need to be reviewed.

• The conclusion in lines 228 and 233 states that Ni could be used for battery production, but this statement is inconsistent with the results, which did not address nickel. Furthermore, there was no mention of what was done with the nickel-enriched solution.

• The conclusion begins with a meaningless sentence.

Some grammatical flaws such as:

Line 42: diffusion dialysis (DD) is an attractive method that (is) widely used in the treatment of acidic wastewater

Line 46: The acid group anions can (be) freely transported through the AEM, while the metallic cations were denied (rejected) by the positive fixed charges.

Repetitive sentences, such as lines 80-90, use simplistic and basic language.

Author Response

Thanks for you comments.

Reviewer 3 Report

The work is of interest to readers of Membranes. However, in order to be satisfied with the publication, it is necessary to finalize the article and make the changes below.

Substantive remarks

1. The introduction is poorly written. The choice of diffusion dialysis and electrodialysis for the separation of the proposed mixture is not justified. Specific examples of successful application of these methods are not given. Which membranes are commonly used? Which ones are better?

2. Table 1 - It is necessary to justify why such composition and concentrations of substances were chosen. If the composition was taken from literature, references should be given.

3. The choice of membranes used in the study is not substantiated.

Formal remarks

1. Remove the abbreviation (DD and ED) in the title.

2. line 142 - Correct to Figure 2.

3. Improve the quality of the figures.

4. Figures 5 and 6 - Denote what the concentrations of H2SO4 on the left and right refer to.

5. The description for Figures 5 and 6 uses g/L and the figures use mol/L. Reduce to the same unit of measure.

6. lines 223-224 - Remove the sentence from the pattern.

7. I recommend that the authors add keywords.

Minor edits to the English language in terms of syntactical errors and punctuation are required.

Author Response

Thanks for you comments.

Round 2

Reviewer 1 Report

the dilution of ED was filled in the dialysate cell.” – maybe diluate? but not “dilution of ED”.

 “The wastewater from battery recycling industry was used as the feed water of this work,” – please, precise: is it the feed for the DD module?

 “The flow rate of dilute chamber, concentrate chamber and electrode chamber was controlled by 800, 1200, 800 L/h, respectively.” – the flow rate was controlled by 800 L/h? It does not sound good. Please, calculate from these values the linear velocity of the solutions (in e.g. cm/s) – this quantity is also important.

 Fig.5 “dilution, concentration” -> e.g. diluate, concentrate.

Please, calculate what are the losses of acid and salt during the proposed treatment (after each step, and the total losses).

“..and the tunneling mechanism [21],” – wrong reference; H. Strathmann described this mechanism.

 Fig 1b – the use of H2O as an inlet liquid makes no sense. Fig. 1b should be correlated (more or less to Fig. 4).

“the higher flow rate would shorten the retention time for Ni2+ to form complex ions and reduce the opportunity for Ni2+ to diffuse across the membrane [22, 31].” – in these references, there is nothing about Ni2+ and its complex ions.  Please, explain this point.

What were the flows of solutions shown in Fig. 4 which led to the results shown in Figs. 5, 6, and 7? Would it be possible to improve the final results by changing these flows?

 The quality of drawings should be improved (now they are blurry).

Still, English should be improved.

Reviewer 2 Report

Thank you for your efforts in making the requested modifications.

From my perspective, I still notice that the figures have low-quality borders. However, I believe it would be best to bring this issue to the attention of the colleagues in the editorial office. They will be able to determine whether this will have an impact on the final quality of the published paper.

Author Response

Thanks for your comment, the pictures were all high resolution in the docx file.

Reviewer 3 Report

I recommend the article for publication, as all comments have been taken into account.

Author Response

Thanks for your comments.

Round 3

Reviewer 1 Report

Please, introduce this information into the manuscript:

 Point 3: “The flow rate of dilute chamber, concentrate chamber and electrode chamber was
controlled by 800, 1200, 800 L/h, respectively.” – the flow rate was controlled by 800 L/h? It does not sound good. Please, calculate from these values the linear velocity of the solutions (in e.g. cm/s) – this quantity is also important.

Response 3: Thanks for your comment. This was the flow rate of the second stage ED, and the linear velocity was 1.32 cm/s and 1.98 cm/s. In the first stage of ED, the flow rate of dilute chamber and concentrate chamber was 1600 L/h and 2400L/h, and the linear velocity was 1.51 cm/s and 2.27 cm/s.”

Point 5: Please, calculate what are the losses of acid and salt during the proposed treatment (after each step, and the total losses).
Response 5: Thanks for your comment. The losses of acid and salt were responded in the first question of the former response. (In DD process, the loss of acid was 4.13 kg H2SO4/h and the loss of salt was 0.99 kg NiSO4/h. In ED process, the loss of acid was 0.82 kg H2SO4/h.)

Point 8: the higher flow rate would shorten the retention time for Ni2+ to form complex ions and reduce the opportunity for Ni2+ to diffuse across the membrane [22, 31].” – in these references, there is nothing about Ni2+ and its complex ions. Please, explain this point.
Response 8: Thanks for your comment. In the manuscript, the references are [23, 32] instead of [22,31], and in these references, the complex ions were other metals, such as [PbCl3]-, [Pb(OH)3]-, (TiF6)2-, etc. Therefore, Ni2+ would form NiSO4 and [Ni(SO4)2]2-, similarly.
(Zhu, R. S.; Gong, Z. Q.; Jiang, H. Y. A study of the chloride and sulfate complexes of nickel[J]. Journal of Central-South Insititute of Mining and Metallurgy, 1983, 118.)

 Point 9: What were the flows of solutions shown in Fig. 4 which led to the results shown in Figs. 5, 6, and 7? Would it be possible to improve the final results by changing these flows?
Response 9: Thanks for your comment. The flow rates of the dilution and concentration in the first stage ED were 1600 L/h and 2400 L/h, in the second stage ED were 800 L/h and 1200 L/h. In ED process, the effect of flow rates was no need to be investigated, because the purpose of ED process was to obtain the acid production of 150 g/L H2SO4. The parameters such as flow rates, current density and voltage were no need for exploration in this work.

English should be improved.

 „the flow rates of dilute chamber and concentrate chamber were controlled by 1600 and 2400 L/h”

- the flow rates of the dilution chamber and concentrate chamber were maintained at 1600 and 2400 l/h.

It should be corrected:

„In the second stage ED, the flow rates were controlled by 800 and 1200 L/h.”

„The flow rate ratio of water to feed (W/F) was controlled by 1:1, …”

„the feed flow rate was controlled by 300 L/h ..”